# Trans-Brachial TAVI in a Patient with Aortic Isthmus Stenosis: A Case Report

**DOI:** 10.3390/jcm13020308

**Published:** 2024-01-05

**Authors:** Mohammed Saad, Abdelrahman Elhakim, Rene Rusch, Rouven Berndt, Bernd Panholzer, Georg Lutter, Derk Frank

**Affiliations:** 1Cardiology Department, Schleswig-Holstein University Hospital-Kiel, Arnold-Heller-Street 3, 24105 Kiel, Germany; aelhakim@schoen-klinik.de (A.E.); derk.frank@uksh.de (D.F.); 2Vascular Surgery Department, Schleswig-Holstein University Hospital-Kiel, Arnold-Heller-Street 3, 24105 Kiel, Germany; rene.rusch@uksh.de (R.R.); rouven.berndt@uksh.de (R.B.); 3Cardiothoracic Surgery Department, Schleswig-Holstein University Hospital-Kiel, Arnold-Heller-Street 3, 24105 Kiel, Germany; bernd.panholzer@uksh.de (B.P.); georg.lutter@uksh.de (G.L.)

**Keywords:** aortic valve stenosis, aortic isthmus stenosis, transcatheter aortic valve implantation, brachial artery

## Abstract

Background: TAVI indications expand not only to low-risk patients but also to patients with a more complex anatomy and comorbidities. Transfemoral retrograde access is recognized as the first preferred approach according to the current guidelines. However, this approach is not suitable in up to 10–15% of patients, for whom an alternative non-femoral access route is required. Case Presentation: An 83-year-old male patient with known aortic isthmus stenosis presented with severe symptomatic aortic stenosis. Computed tomography revealed a subtotal isthmus stenosis, directly after left subclavian artery origin, with many collaterals extending toward the axillary and subclavian arteries. Duplex ultrasound verified the proximal diameter of the left brachial artery to be 5.5 mm. A successful surgical cutdown trans-brachial TAVI with an Evolut prosthetic valve with a size of 29 mm was performed. On the fourth postoperative day, the patient was discharged, and the three-month follow-up was uneventful. Conclusion: In patients with aortic isthmus stenosis, the brachial artery could be a feasible alternative, as a less invasive access site, which can be determined after careful assessment of the vessel diameter. More data are required to evaluate the safety and efficacy of this access route and to achieve more technical improvements to increase operator familiarity with it.

## 1. Introduction

TAVI indications expand not only to low-risk patients but also to patients with a more complex anatomy and comorbidities. Transfemoral retrograde access is recognized as the first preferred approach according to current guidelines because of its minimal invasiveness and the possibility of performing the procedure under conscious sedation/local anesthesia without intubation. Planning a successful procedure through this route is important to identify potential challenges, such as those pertaining to vessel diameter, tortuosity, aortic arch calcifications, the presence of aneurysms, or thrombotic appositions.

However, in up to 10–15% of patients, this approach is not suitable, for whom an alternative non-femoral access route is required [1].

Several alternative access routes for antegrade or retrograde TAVI procedures have been described, namely trans-axillary, trans-aortic, trans-apical, trans-carotid, trans-septal, and trans-caval routes.

Direct transaortic TAVI access can be achieved using a short TAVI delivery system through the direct trans-aortic TAVI through an upper mini-sternotomy or the right antero-lateral mini-thoracotomy at the second intercostal space.

Limitations include the presence of a patent venous coronary artery bypass graft with proximal anastomosis on the ascending aorta at risk of damage, very short ascending aorta, porcelain aorta, and thorax deformities [2].

Transapical access can be achieved through left antero-lateral mini-thoracotomy at the fifth intercostal space. It represents the historical alternative to the trans-femoral TAVI. Limitations are the presence of apical thrombus and severely reduced left ventricular function.

According to data derived from the German GARY registry, trans-apical access is an independent predictor of one year mortality in TAVI patients, and myocardial scar increases the risk of bleeding, apical aneurysm formation, ventricular rupture, and late arrhythmias [3].

In the first human case of TAVI and in the early phase, an antegrade transseptal approach was utilized. The challenges of this approach include the need for atrial septal crossing, procedural complexity, and potential for injury to the mitral valve apparatus, leading it to being supplanted by other approaches [4]. These challenges have led to this approach becoming non-attractive for interventional cardiologists in comparison to other easier implantation techniques.

The trans-carotid (TC) approach can be considered an alternative access route. The left one is preferred given the straight pathway to the aortic valve. The major limitation of this approach is the increased risk of embolization [5].

The trans-caval approach is considered in patients not qualifying for any other vascular access as the last resort. Nonetheless, life threatening bleedings and retroperitoneal hematomas can happen, necessitating an implant of an aortic covered stent [6].

Among the alternative non-femoral routes, subclavian and axillary entry are the most preferred access routes by virtue of producing better clinical results and being less invasive [7]. Trans-axillary access has received increasing recognition due to its proximity and relatively straight course from the axillary artery to the aortic annulus, in addition to being rarely affected by significant atherosclerosis. The proximal third of the axillary artery represents the ideal target for both surgical and percutaneous approaches, and it is considered the second option when trans-femoral TAVI is not feasible [8]. However, the media of the subclavian and axillary arteries are of the elastic type, while the medium of the femoral artery is of the muscular type, having a thicker and more fibrous adventitia. These characteristics predispose the axillary artery access route to vascular complications, such as bleeding, higher failure rate of the vascular closure device than the femoral one, and of ruptures or dissections [7].

However, it is of paramount importance to emphasize that the choice of alternative access and procedural outcomes should be crucially selected according to appropriate patient selection, center experience, and the available setting. 

In this report, we present a case of successful trans-brachial TAVI performed by surgical cutdown due to aortic isthmus stenosis.

## 2. Materials and Methods: Case Presentation

An 83-year-old male patient with a medical history of hypertension, chronic renal disease, and known aortic isthmus stenosis presented to the emergency ward with a complaint of New York Heart Association class III dyspnea. A physical examination revealed a 5/6 systolic murmur at the right upper sternal border and diminished bilateral femoral artery pulses. Electrocardiography showed sinus rhythm. Transthoracic echocardiography revealed degenerative severe aortic stenosis with a peak/mean gradient of 65/45 mm Hg and an aortic valve area of 0.85 cm^2^. The left ventricular ejection fraction was 55%. Coronary angiography demonstrated noncritical coronary artery disease. Electrocardiogram (ECG)-gated multi-slice computed tomography revealed a subtotal isthmus stenosis, directly after left subclavian artery origin, with many collaterals extending toward the axillary and subclavian arteries (Figure 1).

The aortic valve had a trileaflet anatomy and an asymmetric calcification of the non-coronary cusp (Appendix A), with an annulus perimeter of 79.7 mm and an area of 492.8 cm^2^. The angle between the aorta and left subclavian artery was appropriate for TAVI. The diameter of the left subclavian artery was 9.5 mm, and Duplex ultrasound verified the proximal diameter of the left brachial artery to be 5.5 mm.

After an evaluation by the heart team, TAVI was preferred (EuroSCORE II: 10.6%; STS score mortality: 10.9%). A TAVI procedure using the transfemoral and trans-caval approaches was excluded due to aortic isthmus stenosis. Due to the large size of the brachial artery, we found that the trans-brachial approach could be a feasible and safe distal extension rather than the trans-axillary approach. The decision not to use trans-subclavian and trans-axillary approaches was made due to aortic collaterals to both arteries. Access through both arteries carries the risk of collateral injury. The brachial artery was selected with the consensus of the heart team after extensive meticulous planning of the procedure and after obtaining the patient’s consent.

The mid-portion of the left brachial artery was explored under general anesthesia by surgical cutdown and encircled with soft rubber vessel loops (Figure 2). Attention is required to avoid damaging the surrounding structures. A double horizontal 5/0 polypropylene purse string was performed, and the artery was punctured in the center of the purse string (Appendix A).

After the intravenous administration of heparin (aiming to achieve an activated clotting time > 250 s), a 6F sheath, followed by a 10 Fr sheath, was placed over a soft J-tipped 0.035 wire using the Seldinger technique. A pigtail catheter was advanced through the aortic root via the right femoral artery to determine the level of the TAVI valve inside the native aortic valve and to monitor blood pressure. Temporary pacing was performed by means of entry through the right femoral vein. Then, a 0.035-inch soft and flat-tipped guidewire was advanced through the stenotic aortic valve via an Amplatz 1 catheter. A pigtail catheter was advanced over the guidewire. Then, the guidewire was switched for a 0.035-inch extra-stiff wire (Safari XS) positioned in the apex of the left ventricle. Subsequently, a 14F sheath was passed to the ascending aorta via the Safari guidewire (Figure 3) (Appendix A).

Of note, we did not continue to push the sheath to the end. From this point, the procedure followed the same technique used for the TF route. After balloon valvuloplasty (Appendix A) a Medtronic Evolut PRO+ prosthetic valve with a size of 29 mm (Medtronic, Minneapolis, MN, USA) was inserted into the ascending aorta (Appendix A). After confirmation of the position of the Evolut valve (Appendix A), it was implanted with a pacing rate of 120/min with the cusp-overlap technique (Appendix A). There was no paravalvular or valvular regurgitation on the control aortography (Figure 4) (Appendix A).

At the end of the procedure, hemostasis of the brachial artery was achieved by tightening the purse-string sutures, and the skin layers were closed surgically without complication. The patient was discharged after 4 days. The three-month follow-up was uneventful.

## 3. Discussion

We report a case of successful trans-brachial TAVI performed by surgical cutdown dissection due to aortic isthmus stenosis.

The brachial artery is the major artery of the upper extremity and is the extension of the axillary artery, starting at the lower margin of the teres major muscle. The brachial artery courses along the ventral surface of the arm and gives rise to multiple smaller branching arteries (including the deep brachial artery, the superior ulnar collateral artery, and the inferior ulnar collateral artery) before reaching the cubital fossa. In the cubital fossa, it divides into its terminal branches: the radial and ulnar arteries of the forearm [9].

The average diameter of the brachial artery is 4 to 5 mm, but it can vary according to age, sex, and flow rate [9]. The diameter of the left brachial artery of our patient was 5.5 mm. Of note, the brachial artery diameter is larger in particular patients, such as those with aortic isthmus stenosis, due to remodeling and an increase in arterial pressure in the upper limb compared to that of the lower limb. To avoid collateral injury, the left brachial artery was selected as the entry site.

Since the first TBr-TAVI case performed by Bruschi G. et al. in 2018, few cases have been successfully implanted through TBr access [10].

Arslan Ş. et al. performed TBr-TAVI with SAPIEN 3 (Edwards, Irvine, CA, USA) with a size of 26 mm in a patient with bilateral total occlusion of the iliac arteries [11].

Topalo R. et al. performed another case of TBr-TAVI with CoreValve Evolut R 34 mm (Medtronic, Minneapolis, MN, USA) due to extreme tortuosity of both pelvic arteries [12].

To the best of our knowledge, this is the first case report for trans-brachial TAVI in a patient with aortic isthmus stenosis.

The potential benefits of trans-brachial access over other non-TF options are the maintenance of left ventricular integrity (in contrast to transapical access), the absence of chest opening (in comparison to direct aortic access), relatively easy access, minimal tortuosity and calcification, and the avoidance of the preparation of an artery in the otherwise very complex axillary and subclavian regions; these benefits therefore decrease the chances of iatrogenic injury, reduce procedural complexity and invasiveness with a quick recovery after the procedure, and offer a TAVI approach in patients with aortoiliac occlusive disease.

The trans-carotid approach could be a feasible alternative access site in this case, but we do not have experience with this access site. Of note, the result of the French Transcarotid TAVR prospective multicenter registry, including 314 patients treated with the Edwards Sapien 3 device, demonstrates promising data regarding the safety and effectiveness of this approach. The 30-day mortality was 3.2%, rate of major bleeding was 4.1%, and stroke or transient ischemic attack was 1.6% [13]. On the other hand, the result of a large meta-analysis of TC-TAVI, including 1374 TC patients, demonstrates an increased risk of 30-day mortality, and a subgroup analysis of the two propensity score-matched studies found a statistically increased risk of 30-day neurovascular complications (RR, 1.61, 95% CI, 1.02–2.55, *p* = 0.040) [14].

Thus, it is of paramount importance to emphasize that the choice of alternative access and procedural outcomes should be crucially selected according to appropriate patient selection, center experience, and the available setting. 

In addition, the left trans-brachial access site offers better and easier coaxial orientation over the trans-carotid access site. Moreover, the brachial artery has a large diameter, up to 55 mm, particularly in patients such as those with artic isthmus stenosis, as the pressure in the upper limb is higher than in the lower limb.

Interestingly, a recently published case report demonstrates the feasibility of the single-access Impella CP (Abiomed, Danvers, MA, USA) technique through a left brachial artery cutdown approach without access-related complications [15].

The suitable anatomy and positive results when using a large bore sheath with brachial artery access influenced the decision to choose the left trans-brachial artery as the access site for TAVI. A possible limitation, however, is the vessel diameter and assumed risk of vascular complications, such as ischemia and bleeding.

Of note, we decided against angioplasty or the stenting of asymptomatic isthmus stenosis to facilitate transfemoral TAVI.

Technical considerations regarding this access site are as follows:-A preoperative high-resolution computed tomography scan is mandatory to determine the brachial artery suitability, the vessel diameter, the degree of tortuosity, the relationship with side branches, and the presence and extension of calcifications; to plan the procedure; to select the device; to minimize complications; and to improve the intervention’s outcomes.-The left brachial artery is preferred, as it allows for better coaxial orientation, decreases the chance of carotid compromise, and can be advantageous in right-handed patients.-The left brachial artery was reached in this case by surgical cutdown.-An extra-stiff guidewire such as Lunderquist can be used if the Safari wire does not offer the required support.-It is important not to continue pushing the delivery sheath to the end.-The aortic annular angle and the takeoff angulation of the subclavian and innominate arteries with the aortic arch should be considered. An angle >70° between the annular plane and the left subclavian (i.e., “horizontal aorta”) or >30° between the annular plane and the right subclavian horizontal axis typically means a contraindication due to difficulties in achieving coaxiality [8].-A type 1 arch (with all three great vessels originating from the transverse arch) also represents a reason to avoid a right-sided approach, especially if the innominate artery arises distal on the arch [7].-In patients with a patent left internal mammary artery (LIMA) coronary bypass graft, a non-significant atherosclerotic disease proximal to or at the ostium of the LIMA and a minimal vessel diameter of 7–8 mm are essential in order to prevent myocardial ischemia [8].-A major limitation of this approach is the diameter of the brachial artery.-To the best of our knowledge, we were not able to compare self-expanding valves with balloon-expandable valves via this approach as we did not have enough literature data to make the comparison in this case report. In addition, there are case reports for both systems. We preferred a self-expandable valve as it does not expand the sheath, hence minimizing the possibility of vascular injury. However, a balloon-expandable valve has a steerable sheath and could more easily overcome the coaxial and angle challenges.

We hereby emphasize that the trans-brachial TAVI approach is feasible, with some technical considerations, as an alternative less invasive access site in patients who are not suitable for transfemoral TAVI. However, many series of cases are required to evaluate the safety and feasibility of this access site and to compare the TBr approach with outcomes of TF and other non-TF approaches.

## 4. Conclusions

The brachial artery is a feasible alternative and less invasive access site, particularly for patients who are not suitable for TAVI through other entry sites, though this decision should be made with technical considerations. The choice of alternative access and procedural outcomes should be crucially selected according to appropriate patient selection, center experience, and the available setting. 

As the indications for transcatheter valve implantation expand to patients with a more complex anatomy and comorbidities, many series of cases are required to evaluate the safety and efficacy of this access route and to achieve more technical improvements to increase operator familiarity with it.

## Figures and Tables

**Figure 1 jcm-13-00308-f001:**
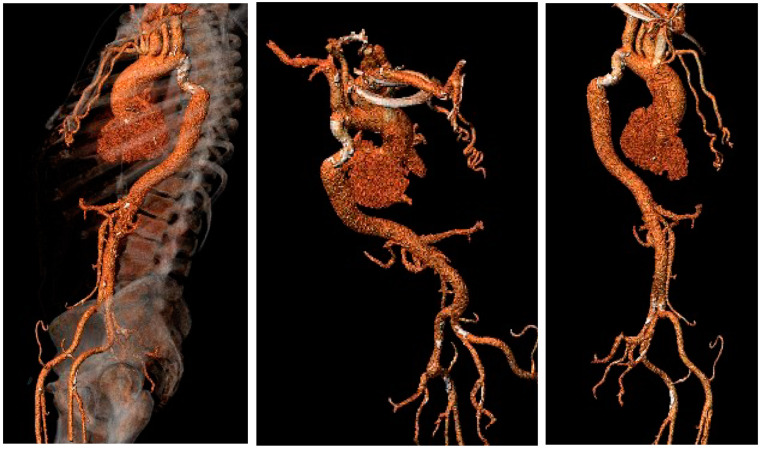
Aortic computed tomography revealed a subtotal isthmus stenosis, directly after left subclavian artery origin, with multiple thoracic collaterals.

**Figure 2 jcm-13-00308-f002:**
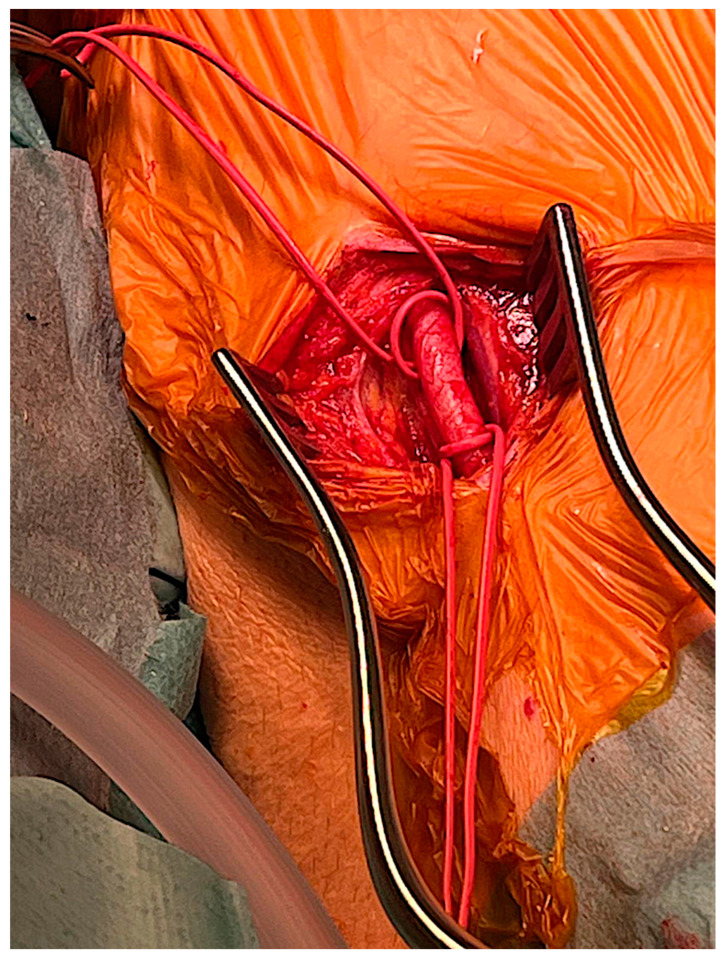
The mid-portion of the left brachial artery was explored by surgical cutdown.

**Figure 3 jcm-13-00308-f003:**
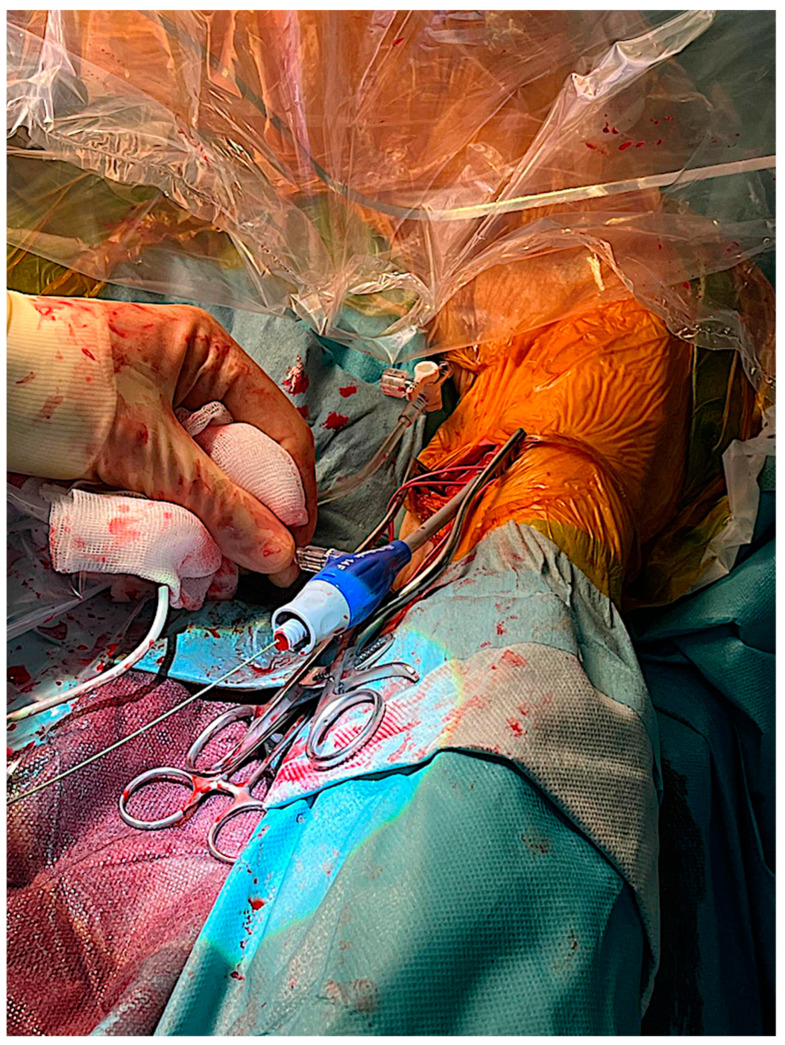
The 14F sheath was passed to the ascending aorta.

**Figure 4 jcm-13-00308-f004:**
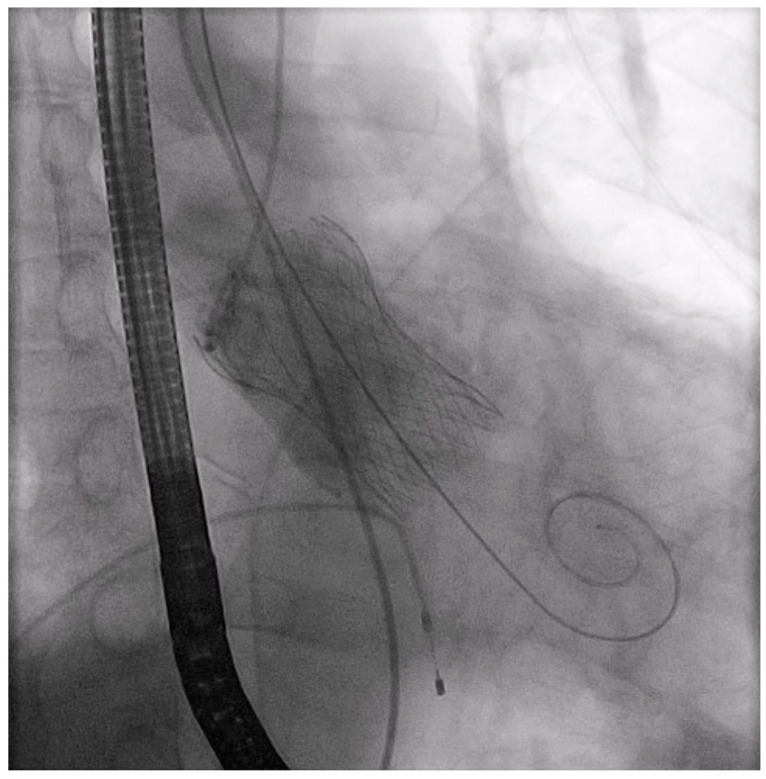
Successful implantation of Evolut valve of 29 mm.

## Data Availability

All data related to the case are available on request. The paper is not under consideration elsewhere; none of the paper’s contents have been previously published; all authors have read and approved the manuscript; the authors take full responsibility for the content of the publication. Declaration of Generative AI and AI assisted technologies in the writing process. The authors of this article disclose that during the preparation of this work, they did not use generative AI and AI-assisted technologies in the writing process of this manuscript. The authors have reviewed and edited this manuscript.

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
