# Peer review of "Trans-Brachial TAVI in a Patient with Aortic Isthmus Stenosis: A Case Report"

_jcm, 2024, doi:10.3390/jcm13020308_

Round 1
Reviewer 1 Report
Comments and Suggestions for Authors
The paper is well written with only minor corrections to perform. The provided figures and video are of great quality and add strength to the quality of the paper.
The manuscript could benefit from the following comments:
Discussion
- Line 194, do the Authors mean a “Lunderquist” guidewire instead of a “Linderquist”?
- Line 194, this point needs further clarification. The Authors managed to use a Safare guidewire during the procedure (Lines 143-145), why do they mention that it did not offer the needed support?
Language comments in the other comment box.
Comments on the Quality of English LanguageLanguage
- The same verbal tense should be consistent throughout the paper. Tense revision is warranted.
- “The patient discharged after” should be corrected to “The patient was discharged after”
Author Response
Point by point response letter
Reviewer (1): The paper is well written with only minor corrections to perform. The provided figures and video are of great quality and add strength to the quality of the paper.
The manuscript could benefit from the following comments:
Discussion
- Line 194, do the Authors mean a “Lunderquist” guidewire instead of a “Linderquist”?
I corrected it to Lunderquist, thank you.
- Line 194, this point needs further clarification. The Authors managed to use a Safare guidewire during the procedure (Lines 143-145), why do they mention that it did not offer the needed support?
I corrected it as follwoe: An extra stiff guidewire like Lunderquist can be used if the Safari wire do not offer the needed support.
Language comments in the other comment box.
Comments on the Quality of English Language
Language
- The same verbal tense should be consistent throughout the paper. Tense revision is warranted.
- “The patient discharged after” should be corrected to “The patient was discharged after”
I revised the manuscript and applied it for an extensive english editing service.

Reviewer 2 Report
Comments and Suggestions for Authors
The authors describe an interesting case of left transbrachial TAVR via an opening cut down approach in a patient with high surgical risk and aortic isthmus stenosis.
This is overall a very good case since only several cases like this have been performed.
Several comments:
1) Why did the operators decide against trans carotid or the right subclavian/axillary approach? The trans carotid approach has been recently advocated for as one of the safer access sites for alternative access TAVR (definitely more experience than transbrachial). You have to have a very good reason to not using an acceptable alternative access before going through a fairly experimental access like the brachial.
2) What was the maximal diameter across the isthmus stenosis. Has there been a discussion regarding stenting the isthmus stenosis/ ballooning it and going transfemoral?
3) You can also include in the discussion that there has been reports of impella CP placement via brachial access 14F which may add some credibility to this approach for patients with no alternative access and suitable brachial artery size.
4) Are there better images showing the collaterals and their locations?
5) How do you compare using self-expanding vs balloon expandable valves via this approach, would you recommend self-expandable valve as they dont expand the sheath hence minimizing the possibility of vascular injury, include this in the discussion.
Comments on the Quality of English LanguageRevise the manuscript a couple more times for typos and grammar.
Author Response
Point by point response letter
Reviewer (2): The authors describe an interesting case of left transbrachial TAVR via an opening cut down approach in a patient with high surgical risk and aortic isthmus stenosis.
This is overall a very good case since only several cases like this have been performed.
Several comments:
- Why did the operators decide against trans carotid or the right subclavian/axillary approach? The trans carotid approach has been recently advocated for as one of the safer access sites for alternative access TAVR (definitely more experience than transbrachial). You have to have a very good reason to not using an acceptable alternative access before going through a fairly experimental access like the brachial.
- The left brachial artery is preferred as it allows better easier coaxial orientation, decreases the chance of carotid compromise, is the extension of axillary artery and can be advantageous in right-handed patients.
- In addition, we did a case series of trans-brachial EVAR and have a good experience with large bore sheath via brachial artery access.
- Moreover, the brachial artery has large diameter up to 55 mm in particular patients like artic isthmus stenosis as the pressure in the upper limb is higher than the lower limb.
- The aortic annular angle and the takeoff angulation of the subclavian and innominate artery with the aortic arch has been considered and were suitable.
- The left subclavian artery was not retroflexed towards the descending aorta, or with a steep subclavian to arch angulation (> 80°), thus suitable anatomy for doing this procedure.
- Left subclavian and axillary arteries were excluded as access site to avoid collateral damage and injury.
- Right subclavian artery does not allow easier coaxial orientation as left one but can be used as an alternative access site.
- Trans-carotid approach could be also possible option in this case, but we do not have experience with this access site. In addition, left trans-brachial access site offer better easier coaxial orientation over trans-Carotis. The suitable anatomy and good experience with this access site has influenced the decision for left trans-brachial artery as access site for TAVI.
- What was the maximal diameter across the isthmus stenosis. Has there been a discussion regarding stenting the isthmus stenosis/ ballooning it and going transfemoral?
We did not measure it and we did not treat it with balloon angioplasty as the patient was asymptomatic regarding artic isthmus stenosis.
- You can also include in the discussion that there has been reports of impella CP placement via brachial access 14F which may add some credibility to this approach for patients with no alternative access and suitable brachial artery size.
I found in the literature a case reports of IMPELLA 2.5L through brachial artery. However, Impella 2.5 L is 13.5 f and is not available anymore and is replaced by Impella CP. We mentioned, however, case reports of trans brachial TAVI.
Could the reviewer provide me please with this case report to add it?
- Are there better images showing the collaterals and their locations?
We did not have a good image regarding collaterals and their location. The vascular surgeon advised against left trans axillary and trans subclavian artery as access site for TAVI to avoid injury and damage of collaterals in this particular patient with aortic isthmus stenosis.
- How do you compare using self-expanding vs balloon expandable valves via this approach, would you recommend self-expandable valve as they dont expand the sheath hence minimizing the possibility of vascular injury, include this in the discussion?
It could not be able to compare self-expanding vs balloon expandable valves via this approach as we do not have enough data to confront it with this case report. In addition, there are case reports of both systems. We preferred self-expandable valve as it does not expand the sheath hence minimizing the possibility of vascular injury. On the other hand, balloon expandable valve has a steerable sheath and could be easier to overcome the coaxial and angels’ challenges.
Comments on the Quality of English Language
Revise the manuscript a couple more times for typos and grammar.
I revised the manuscript and applied it for an extensive English editing service.
Round 2
Reviewer 2 Report
Comments and Suggestions for Authors
In the introduction in the portion about alternative access discussion, it is very important to emphasize that the alternative access of choice should be guided by center experiance.
In the discussion about the trancorotid access, french registry actually found that risk of stroke is compatible and even lower than transfemoral while a larger meta-analysis found slightly increased risk so it is all guided by center experiance and appropriate patient selection (PMID: 30772290, PMID: 34124210). Include those. Additonally, the transcaval access, i wouldnt say it is the last resort, a lot of center especially in the US would have chosen transcaval over the transbrachial so modify the language there suggesting that centers who have the equipment and proficient at transcaval may prefer that access.
- Case report in which impella CP sheath was used: PMID: 37551678
Author Response
Dear Reviewer
Thank you very much for your valuable comments. I change the main document according to your comments as follow:
In the introduction in the portion about alternative access discussion, it is very important to emphasize that the alternative access of choice should be guided by center experiance.
I added this: However, it is paramount important to emphasize that the choice of alternative access and procedural outcomes should be crucially selected according to appropriate patient selection, center experience and the available setting.
In the discussion about the trancorotid access, french registry actually found that risk of stroke is compatible and even lower than transfemoral while a larger meta-analysis found slightly increased risk so it is all guided by center experiance and appropriate patient selection (PMID: 30772290, PMID: 34124210). Include those. Additonally, the transcaval access, i wouldnt say it is the last resort, a lot of center especially in the US would have chosen transcaval over the transbrachial so modify the language there suggesting that centers who have the equipment and proficient at transcaval may prefer that access.
Of note, the result of the French Transcarotid TAVR prospective multicenter registry, included 314 patients treated with Edwards Sapien 3 device, demonstrates promising data regarding the safety and effectiveness of this approach. The 30-day mortality was 3.2%, rates of major bleeding was 4.1%, and stroke or transient ischemic attack was 1.6% [13]. On the other hand, the result of a large meta-analysis of TC-TAVI, included 1,374 TC patients, demonstrates an increased risk of 30-day mortality, and a subgroup analysis of the two propensity-score matched studies found a statistically increased risk of 30-day neurovascular complications (RR, 1.61, 95% CI, 1.02-2.55, p = 0.040) [14].
Thus, it is paramount important to emphasize that the choice of alternative access and procedural outcomes should be crucially selected according to appropriate patient selection, center experience and the available setting.
Case report in which impella CP sheath was used: PMID: 37551678
I added this: Interestingly, a recently published case report demonstrates the feasibility of the single-access Impella CP (Abiomed, Danvers, MA, USA) technique through a left brachial artery cutdown approach without access related complications [15].